# Double-blinded randomized controlled trial to reveal the effects of Brazilian propolis intake on rheumatoid arthritis disease activity index; BeeDAI

Yoshinari Matsumoto[1,2], Kanae Takahashi[3], Yuko Sugioka[4], Kentaro Inui[5], Tadashi Okano[5], Koji Mandai[5], Yutaro Yamada[5], Ayumi Shintani[3], Tatsuya Koike[1,4]*

1 Search Institute for Bone and Arthritis Disease (SINBAD), Shirahama Foundation for Health and Welfare, Wakayama, Japan, 2 Department of Medical Nutrition, Osaka City University Graduate School of Human Life Science, Osaka, Japan, 3 Department of Medical Statistics, Osaka City University Medical School, Osaka, Japan, 4 Center for Senile Degenerative Disorders (CSDD), Osaka City University Medical School, Osaka, Japan, 5 Departments of Orthopaedic Surgery, Osaka City University Medical School, Osaka, Japan

* tatsuya@med.osaka-cu.ac.jp

**Data Availability Statement:** Since there are no restrictions, we upload the data set as a Supporting Information file.

## Abstract

### Background and aims

Brazilian propolis reportedly contributed to suppressing disease activity in a mouse model of rheumatoid arthritis (RA), suggesting new treatment options using Brazilian propolis. However, only results from animal experiments have been available, and the suppressive effects of Brazilian propolis on disease activity in humans with RA remain unknown. The purpose of this study was to clinically validate how Brazilian propolis intake changes disease activity in RA patients.

### Methods

This study was conducted as a multicenter, double-blinded, randomized, placebo-controlled, parallel-group study of 80 women with RA (median age, 61.5 years; interquartile range, 56.0 to 67.3 years) showing moderate disease activity on Disease Activity Score in 28 joints using erythrocyte sedimentation rate (DAS28-ESR). Test tablets containing Brazilian propolis were used in Group P (40 patients), and Brazilian propolis-free placebo tablets were used as control in Group C (40 patients). Group P received 5 tablets of propolis (508.5 mg of propolis) daily, and Group C received 5 tablets of placebo daily. The intervention lasted 24 weeks, with change in DAS28-ESR set as the primary endpoint. As secondary endpoints, other disease activity assessment (DAS28 using C-reactive protein, simplified disease activity index, clinical disease activity index), ultrasonographic evaluation of synovitis, activities of daily living, quality of life, changes in cytokine levels, and adverse events over the course of the study were also assessed. Data were statistically analyzed by analysis of covariance.

**Funding:** Y.M. received Yamada Research Grant (grant No.249YM) from a commercial source: Yamada Bee Company, Inc. (https://www.3838.com). The funders had a role in preparing and blinding propolis and placebo tablets and in decision to publish but had no further role in study design, data collection and analysis, or preparation of the manuscript.

**Competing interests:** Yoshinari Matsumoto; Grant/research support from: Yamada Research Grant Kanae Takahashi; Grant/research support/Speakers bureau: no COI Kentaro Inui; Grant/research support: Janssen Pharmaceutical K.K., Astellas Pharma Inc., Sanofi K.K., Abbvie GK, Takeda Pharmaceutical Co. Ltd., QOL RD Co. Ltd., Mitsubishi Tanabe Pharma, Ono Pharmaceutical Co. Ltd., Eisai Co., Ltd., Speakers bureau: Daiichi Sankyo Co. Ltd., Mitsubishi Tanabe Pharma, Janssen Pharmaceutical K.K., Astellas Pharma Inc., Takeda Pharmaceutical Co. Ltd., Ono Pharmaceutical Co. Ltd., Abbvie GK, Pfizer Inc., Eisai Co., Ltd., Chugai Pharmaceutical Co., Ltd., Tadashi Okano; Grant/research support: AbbVie, Eisai, Mitsubishi Tanabe Pharma Corporation and Nipponkayaku, Speakers bureau: AbbVie, Asahikasei, Astellas Pharma Inc, Ayumi Pharmaceutical, Bristol-Myers Squibb, Chugai Pharmaceutical, Daiich Sankyo, Eisai, Janssen, Lilly, Mitsubishi Tanabe Pharma Corporation, Novartis Pharma, Ono Pharmaceutical, Pfizer, Sanofi, Takeda Pharmaceutical, Teijin Pharma and UCB Koji Mandai; Speakers bureau: Asahikasei, Eisai Yutaro Yamada; Speakers bureau: Abbvie, Chugai, Eisai, and Mitsubishi Tanabe Ayumi Shintani: Grant/research support: no COI, Speakers bureau: Chugai Pharmaceutical Co., Ltd. Tatsuya Koike; Grant/research support: AbbVie, Astellas Pharma Inc, Bristol-Myers Squibb, Chugai Pharmaceutical, Eisai, Janssen, Lilly, Mitsubishi Tanabe Pharma Corporation, MSD, Ono Pharmaceutical, Pfizer, Roche, Takeda Pharmaceutical, Teijin Pharma, and UCB, Speakers bureau: AbbVie, Astellas Pharma Inc, Bristol-Myers Squibb, Chugai Pharmaceutical, Eisai, Janssen, Lilly, Mitsubishi Tanabe Pharma Corporation, MSD, Ono Pharmaceutical, Pfizer, Roche,Takeda Pharmaceutical, Teijin Pharma, and UCB. This research was supported (in part) by Yamada Research Grant. This does not alter our adherence to PLOS ONE policies on sharing data and materials. There are no patents, products in development or marketed products to declare.

## Results

No significant differences in the primary endpoint were identified between groups (Group P vs Group C, effect: 0.14, 95% confidence interval: -0.21 to 0.49, p = 0.427). Likewise, no significant differences were seen between groups for any secondary endpoints. The adverse event rate during the study period was 28% in Group P and 33% in Group C.

## Conclusions

Brazilian propolis exerted no effects on disease activity in patients with RA.

## Introduction

Rheumatoid arthritis (RA) is an autoimmune disease that has an unknown cause. RA is thought to be triggered by exposure to environmental factors such as smoking and periodontal disease under some genetic background [1]. When an autoimmune reaction begins, joint or systemic inflammation results in joint swelling and pain that progresses to joint destruction and deformity [1]. In RA, osteoclasts are activated by inflammatory cytokines, resulting in bone destruction [2]. The prevalence of RA in adults is about 1% worldwide, with a high incidence in women between 30 and 50 years old [3]. Currently, the main treatments for RA include pharmacotherapy and surgical intervention, with the former as the cornerstone of RA treatment [4]. Medications include conventional synthetic disease-modifying anti-rheumatic drugs (csDMARDs), biological or targeted synthetic DMARDs (b-/ts-DMARDs), and glucocorticoids aimed at reducing inflammation [4]. All the pharmacotherapies show problems such as side effects or high medical cost, especially with the use of b-/ts DMARDs.

Propolis is a sticky, resinous substance produced by bees through the collection of plant buds, sap and pollen that are mixes with bee saliva and beeswax. Brazilian propolis mainly contains cinnamic acid derivatives such as artepirin C, dorpanin and p-cumulanic acid, and also contains other caffeine oil quinacids, such as flavonoids and chlorogenic acid [5, 6]. Various pharmacological properties of Brazilian propolis have been reported, including antioxidant activity [7], improvement in periodontal disease [8] and inhibition of cognitive decline [9]. Brazilian propolis has also been reported to contribute to the suppression of disease activity against arthritis in a mouse model of RA [10]. As a potential mechanism of action, Brazilian propolis has been reported to suppress interleukin (IL)-17 production by inhibiting the phosphorylation of signal transducer and activator of transcription (STAT)3, which is necessary for the process of differentiation of naïve helper T cells into IL-17-producing helper T cells (Th17 cells) [11]. Brazilian propolis may contribute to the suppression of disease activity in patients with RA, and confirmation of such contributions in humans may lead to the development of alternative therapies for RA. However, at present, only the results from animal experiments in the mouse RA model are available, and the suppressive effects of propolis on disease activity in RA patients remain unclear. The present study was therefore designed to clinically validate changes to disease activity resulting from Brazilian propolis in RA patients.

## Subjects and methods

### Patients in the study

This multicenter, double-blinded, randomized, placebo-controlled, parallel-group study was conducted with the approval of Osaka City University Hospital Certified Review Board (approval number; 203966; March 28, 2018). The study was registered as a clinical trial in the University Hospital Medical Information Network (UMIN 000032149) and the Japan Registry

of Clinical Trials (jRCTs051180142). Written informed consent was obtained from all subjects prior to their participation in this study, in accordance with the Declaration of Helsinki. Patients were enrolled between July 3, 2018 and March 14, 2019. First of all, the trial was registered with UMIN Clinical Trials Registry and the registration date was April 7, 2018. However, Japan's Clinical Trials Act went into effect on April 1, 2018, and our research applicable under the category of Specified Clinical Research. Because the study could not be completed within the registration grace period, we again registered our study with jRCTs as Specified Clinical Research. The registration date for jRCT was March 20, 2019. We confirm that all ongoing and related trials for this intervention are registered. This study was conducted at Osaka City University Hospital (Osaka, Japan), Shirahama Hamayu Hospital (Wakayama, Japan), and Kitade Hospital (Wakayama, Japan). All RA patients attending outpatient clinics for these three hospitals were initially considered eligible for inclusion. All participants met the 2010 American College of Rheumatology (ACR)/European League against Rheumatic Diseases (EULAR) classification criteria for RA [12], and eligibility criteria for the study were as follows: 1) age between 30 and 70 years old; 2) female sex; 3) moderate disease activity (MDA) according to the EULAR disease activity classification [13] ($3.2 <$ Disease Activity Score in 28 joints using erythrocyte sedimentation rate (DAS28-ESR) $\leq 5.1$ [14], where DAS28-ESR $< 2.6$ is classified as remission, $2.6 \leq$ DAS28-ESR $< 3.2$ as low disease activity, and DAS28-ESR $> 5.1$ as high disease activity; and 4) history of taking stable doses of b-/ts-DMARDs, csDMARDs, non-steroidal anti-inflammatory drugs (NSAIDs) and/or glucocorticoid ($\leq 10$ mg/day prednisolone-equivalents) for 12 weeks prior to study entry. Exclusion criteria included: 1) inability to attend regular checkups; 2) intake of propolis $\leq 4$ weeks prior to providing study consent; 3) obesity (body mass index (BMI) $\geq 30$ kg/m$^2$); 4) abnormal liver function (aspartate aminotransferase or alanine aminotransferase $> 3$ times the upper limit of normal range); 5) abnormal renal function (blood urea nitrogen $\geq 25$ mg/dl or serum creatinine $\geq 2.0$ mg/dl); 6) current pregnancy or breastfeeding; 7) history of food allergy; or 8) other characteristic deemed as unsuitable in study subjects by the physician responsible for the study. During the 24 weeks of the study, concomitant medications were left unchanged wherever possible, unless disease activity could not be adequately controlled.

## Study design

Among the 82 patients with RA initially considered eligible, 80 patients were randomized by the research electronic data capture (REDCap; https://redcap.med.osaka-cu.ac.jp/redcap/redcap_v9.5.30/index.php?pid=830) system for computerized random assignment to a propolis administration group (Group P; receiving a total of 5 tablets containing 101.7 mg of propolis extract powder per tablet, representing 508.5 mg/day for 24 weeks, n = 40) or a control group (Group C; 5 placebo tablets daily for 24 weeks, n = 40).

The test and placebo tablets were manufactured at Yamada Bee Company, Inc. (Okayama, Japan), then assembled as a set providing a 3-month supply for one person at the Yamada Bee Company, Inc. The propolis powder (lot. LY008), standardized to contain 8.0% artepillin C and 0.14% culifolin was obtained from Yamada Bee Company, Inc. Test food allocation codes were then randomly assigned to the entire patient cohort at the Yamada Bee Company, Inc., and assignment tables were sent to REDCap randomization staff only. To ensure the unidentifiability of test foods: 1) all test tablets were coated in white; 2) the test food layout code table was kept only at Yamada Bee Company, Inc. and by REDCap test food allocation and coding staff, until the key was opened; 3) keys were not to be opened until the study was completed and data for each case were fixed, except in cases of emergency evacuation for relevant cases. Test tablets were packaged individually as a daily supply of five tablets, and distributed on the

day they were assigned. The timing of dosing was not specified. Ingestion of more than five tablets per day was prohibited, and study participants were allowed to divide up intake into portions, as long as they were able to take all required tablets during the course of the day. To ensure compliance with the dosage, the patients were instructed that if they did not take five tablets in one day, they should not take them the next day and return the entire bag at their next visit. Sachets were collected at each visit, and any leftover intake was checked and recorded at the time of the visit.

## Clinical assessment

Various surveys were conducted in accordance with the schedule. Blood tests were performed under fasting conditions to exclude the effects of diet at 0, 12, 24, and 36 weeks. Serum was stored at -80˚C until the measurement of various laboratory parameters. The following data were obtained: 1) patient background characteristics of age, lifestyle (smoking, drinking, exercise habits, exercise restrictions, and supplement use), and degree of RA dysfunction according to ACR criteria [15]; 2) intake of the test food at every visit; 3) subjective symptoms and objective observations (DAS28-ESR, DAS28 using C-reactive protein (DAS28-CRP) [16], simplified disease activity index (SDAI) [17], and clinical disease activity index (CDAI)) [17]; 4) height and weight; 5) blood and biochemical data; 6) cytokines and RA-specific blood tests (anti-cyclic citrullinated peptide antibody (CCP), matrix metalloproteinase (MMP)-3, interleukin (IL)-6, IL-17A, IL-10; laboratory testing for these items were performed by LSI Medience Corp, Tokyo, Japan); 7) quality of life (QOL) using the 36-item short-form health survey (SF-36) [18]; 8) activities of daily living (ADL) using the modified health assessment questionnaire (mHAQ) [19]; 9) sonographic examination of synovitis in 7 joints on the side with stronger symptom at baseline (unchanged during the observation period) (2nd and 3rd metacarpophalangeal (MP) joints, 2nd and 3rd proximal interphalangeal (PIP) joints, wrist, and 2nd and 5th metatarsophalangeal (MTP) joints) using both grayscale (GS) (grade 0–3) and power-Doppler (PD) scale (grade 0–3) [20], calculated as the sum of GS and PD scores for the seven joints; and 10) adverse events (mild: treatment-free status; moderate: administration can be continued with some treatment; severe: condition to be stopped or discontinued). Serious adverse events were also assessed for any unfavorable medical event, regardless of the dose, that would result in death, life-threatening status, or a requirement for hospitalization for treatment.

## Study outcomes

The change (∆) in DAS28-ESR from baseline (BL) to 24 weeks (24 w) was considered as the primary outcome of the study. DAS28-ESR values at 12 weeks and 36 weeks were also analyzed as a complementary analysis to the primary outcome. Secondary outcomes were evaluated as: 1) ∆DAS28-CRP, ∆SDAI and ∆CDAI from BL to 24 w, and DAS28-CRP, SDAI and CDAI at 12, 24 and 36 weeks (at 3 time points); 2) ∆SF-36 score from BL to 24 w; 3) ∆ each cytokines from BL to 24 w; 4) ∆mHAQ score from BL to 24 w, and mHAQ score at the 3 time points; 5) ∆sum of GS and PD joint grades from BL to 24 w, and sum of joint grades at the 3 time points; 6) adverse events requiring interruption or discontinuation of the test food up to 24 weeks; 7) serious adverse events up to 36 weeks; and 8) all adverse events up to 36 weeks.

## Statistical analysis

Statistical analysis was performed after all test data became available, with no disclosure of patient allocations. Power calculations were performed before starting the study, with reference to a clinical trial testing the effects of an intervention with a Mediterranean diet for RA patients as a similar study [21]. In that clinical trial, mean ∆DAS28-ESR between control and

intervention groups after 12 weeks was 0.4. In our data from another cohort study of RA patients and non-RA subjects [22], mean ΔDAS28-ESR under standard of care for RA patients with MDA over 1 year was -0.22 and an add-on effect for propolis of -0.5 was assumed. To detect with 80% power by two-sample t-test with σ = 0.78 and α = 0.05, at least 39 patients were needed in each group. A total of 80 patients (40 in each group) was thus set as the number of cases required to provide adequate statistical power for this investigation.

Two-tailed tests and two-tailed 95% confidence intervals (CIs) were used for hypothesis testing at the 5% level of significance. Categorical data are presented as percentages and frequencies, and continuous data are presented as medians and quartile ranges. For efficacy analysis, the full analysis set (FAS) was analyzed, defined as the population of participants not meeting the following conditions: 1) participants who never consumed the test food; 2) participants who were never assessed regarding the primary endpoint after consuming the study food; 3) participants who withdrew consent to provide information; 4) participants with the failure to satisfy major entry criteria; 5) participants with the lack of any data post randomization. A complementary statistical analysis was also performed with the per protocol set (PPS), excluding participants meeting the following conditions: 1) participants with < 70% intake of the test food; 2) participants who did not complete the study; 3) visits with missing data on the primary endpoint; and 4) visits for the primary endpoints collected outside the acceptable date range (±7 days); and 5) participants found to be ineligible after randomization. For missing data caused by 3) and 4), values were completed using last observation carried forward (LOCF) when performing the main analysis. Among randomized cases, the population excluding cases meeting the following criteria was defined as the safety analysis set (SAS): 1) participants who had never been evaluated for safety after consumption of the study food; and FAS exclusion criteria 3). In the main analysis of the primary endpoint, a model in which the objective variable was the ΔDAS28-ESR from 0 to 24 weeks, groups (Groups P and C) were used as the explanatory variables, and the value of DAS28-ESR at 0 weeks was the covariate, with significance for the coefficients between groups determined by analysis of covariance (ANCOVA). The normality of residuals was checked using Normal QQ plots. If the plots suggest that the residuals are not normally distributed, the log transformation of the outcome variable was performed, and the bootstrap CI (number of repetitions set to 1000) was also calculated when the assumption of normality regarding the residuals of the log-transformed model. The following analyses were performed as the complementary analyses. First, mixed-effects model analyses were performed for the primary endpoints to confirm the robustness of results on the LOCF completion of missing values. The covariance structure within patients in the model was unstructured. The objective variable was set as DAS28-ESR at 12 and 24 weeks, and the explanatory variables were set as group (Groups P and C), time point, and group and time point interaction, with the value of DAS28-ESR at 0 weeks as a covariate and subjects as a random effect. Second, a mixed-effects model analysis with DAS28-ESR at 12, 24, and 36 weeks data was also performed. The analyses for secondary endpoints 1, 2, 3, 4 and 5 were also performed in the same way as for the primary endpoint (ANCOVA). For the safety analysis, a frequency table was created for secondary endpoints 6, 7, and 8 in the SAS. Statistical analysis was performed using R version 4.0.0 (R Core Team (2020). R: A language and environment for statistical computing. R Foundation for Statistical Computing, Vienna, Austria. URL https://www.R-project.org/.).

## Results

### Study participants

The flowchart of study participants is shown in Fig 1. In Group P, the study was discontinued in 4 cases and intake of the test food was < 70% in 2 cases, resulting in a PPS of 34 cases. In Group C, intake of the test food was < 70% in 4 cases, resulting a PPS of 36 cases.

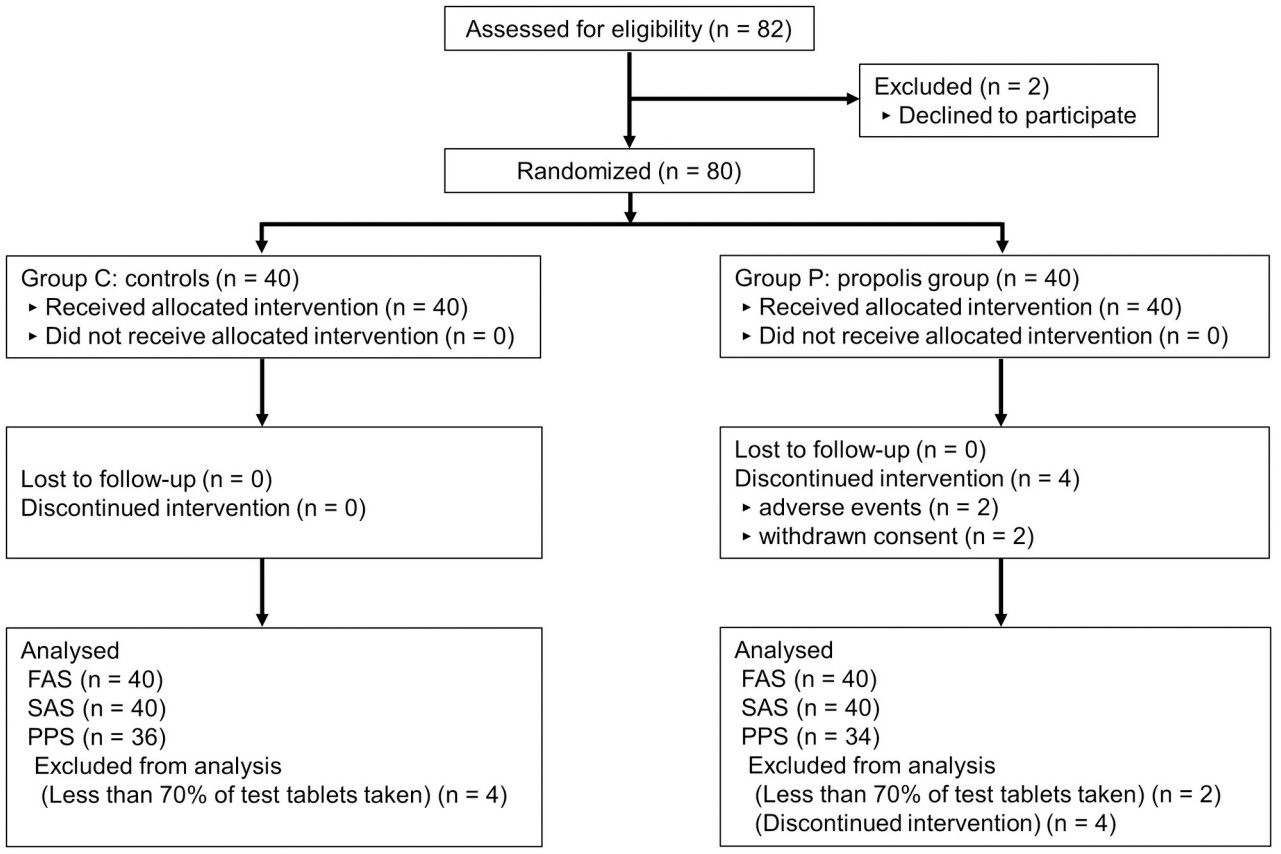

**Fig 1. Study flowchart.** FAS, full analysis set; PPS, per protocol set; SAS, safety analysis set.

## Baseline and demographic characteristics

Demographic and baseline characteristics are presented in Table 1. The median and interquartile range of age of the overall study cohort was 61.5 (56.0, 67.3) years.

**Table 1. Demographic and baseline characteristics of patients.**

|  | Group C (n = 40) | Group P (n = 40) |
|---|---|---|
| Age (years) | 63.5 (56.8, 68.0) | 60.0 (55.8, 65.3) |
| Height (cm) | 153.8 (149.7, 157.5) | 155.8 (152.7, 160.0) |
| Weight (kg) | 51.1 (47.6, 56.6) | 54.1 (46.8, 57.3) |
| BMI (kg/m$^2$) | 21.2 (19.9, 23.8) | 22.1 (19.1, 24.5) |
| Smoking | 8 (20) | 7 (18) |
| Alcohol drinking | 20 (50) | 19 (48) |
| Exercise habits | 11 (28) | 9 (23) |
| Exercise restriction | 0 (0) | 1 (3) |
| Supplement user | 12 (30) | 12 (30) |
| GC user | 6 (15) | 6 (15) |
| GC dose (mg/day) | 2.5 (1.3, 3.8) | 2 (1.3, 2.0) |
| MTX user | 34 (85) | 36 (90) |
| MTX dose (mg/week) | 12 (10.0, 16.0) | 12 (10.0, 12.9) |
| csDMARD other than MTX user | 16 (40) | 21 (53) |
| b-/ts-DMARD user | 10 (25) | 11 (28) |

*(Continued)*

**Table 1.** (Continued)

|  | Group C (n = 40) | Group P (n = 40) |
| --- | --- | --- |
| NSAID user | 18 (45) | 17 (43) |
| Anti-CCP antibody (U/ml) | 189.0 (31.5, 637.0) | 120.00 (13.1, 230.5) |
| MMP-3 (ng/ml) | 40.8 (33.1, 58.8) | 41.7 (30.4, 89.7) |

Data are shown as median (25$^{th}$, 75$^{th}$ percentiles) or number of patients (%).
b-/ts-DMARD; biological or targeted synthetic disease-modifying antirheumatic drug; BMI, body mass index; C, control; CCP, cyclic citrullinated peptide; csDMARD, conventional synthetic disease-modifying anti-rheumatic drug, GC, glucocorticoid; P, propolis; MMP, matrix metalloproteinase; MTX, methotrexate; NSAID, non-steroidal anti-inflammatory drug.

## Primary endpoint and complementary statistical analyses

No significant difference in ΔDAS28-ESR from 0 to 24 weeks was seen between groups in the FAS (Fig 2), and analysis of the PPS showed similar results (data not shown). The result of the

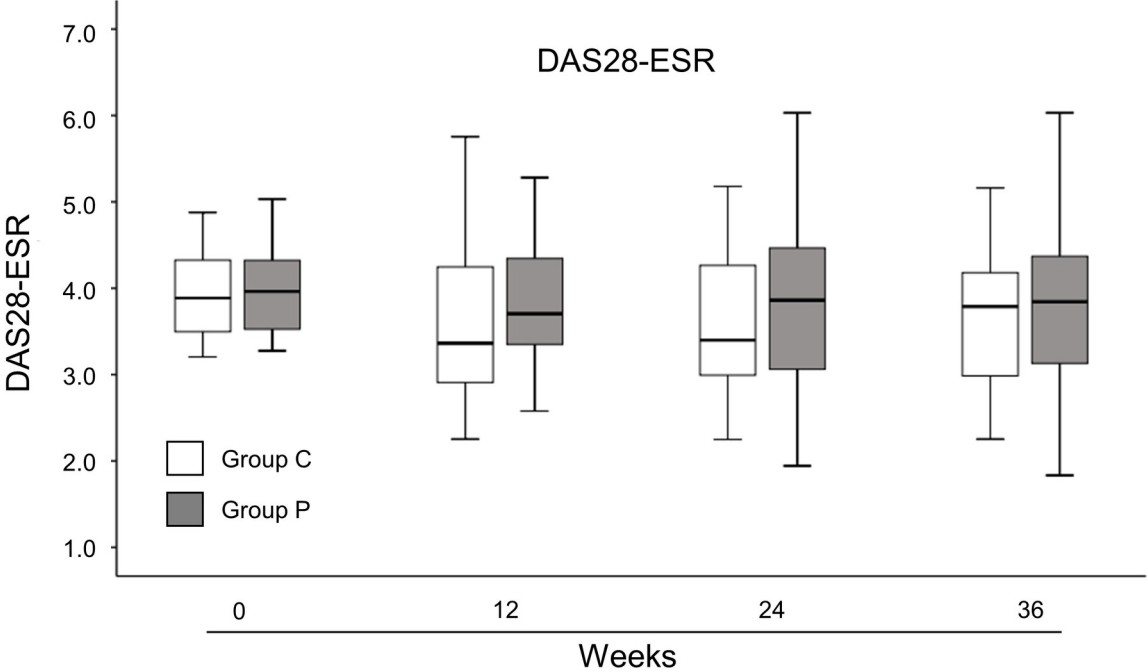

| Change in DAS28-ESR from 0 to 24 weeks (ANCOVA) | p value | Effect (95%CI) |
| --- | --- | --- |
| Group P (vs Group C) | 0.43 | 0.14 (-0.21, 0.49) |
| Mixed-effects model analyses with DAS28-ESR: Group P (vs Group C) |  |  |
| at 12 weeks | 0.14 | 0.24 (-0.08, 0.56) |
| at 24 weeks | 0.38 | 0.14 (-0.18, 0.46) |
| at 36 weeks | 0.78 | 0.05 (-0.27, 0.36) |

**Fig 2. Changes in DAS28-ESR over the course of the study in the FAS.** Data are presented as box-and-whisker plots. White box, Group C; gray box, Group P. ANCOVA; analysis of covariance, C, control; CI, confidence interval; DAS, disease activity score; ESR, erythrocyte sedimentation rate; P, propolis.

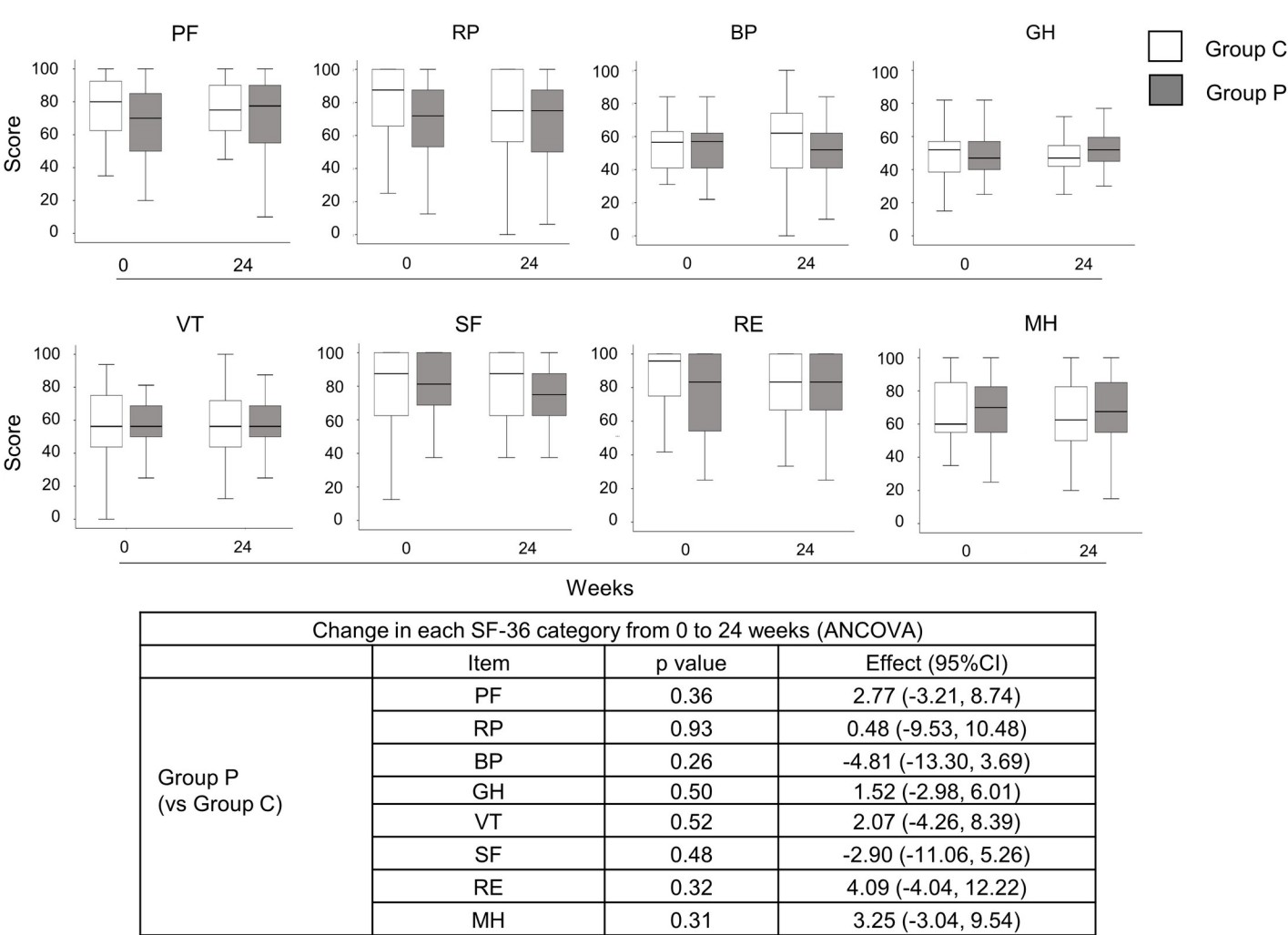

**Fig 3. Changes in SF-36 score over the course of the study in the FAS.** Data are presented as box-and-whisker plots. White box, Group C; gray box, Group P. ANCOVA; analysis of covariance, BP, bodily pain; C, control; CI, confidence interval; GH, general health; MH, mental health; P, propolis; PF, physical functioning; RE, role emotional; RP, role physical; SF, social functioning; SF-36, 36-item short-form health survey; VT, vitality.

mixed-effects model analysis was also described in Fig 2. It showed the main effects of group at each time point. The effect of group was not statistically significant at any time point.

## Secondary endpoints

The same analyses were performed for DAS28-CRP, SDAI, CDAI, and joint sonographic assessment, but no significant differences were apparent between groups (S1–S3 Figs). SF-36 score (Fig 3) and mHAQ score (S4 Fig) also showed no significant differences between groups. The results for cytokines also showed no significant differences (Fig 4). All adverse events that occurred during the study are shown in Table 2. The number of patients with adverse events was 13 (33%) in Group C and 11 (28%) in Group P.

## Discussion

This study comprised a multicenter, double-blind, randomized, controlled trial to examine the effects of propolis on disease activity in RA patients. In the present study, no effect of propolis

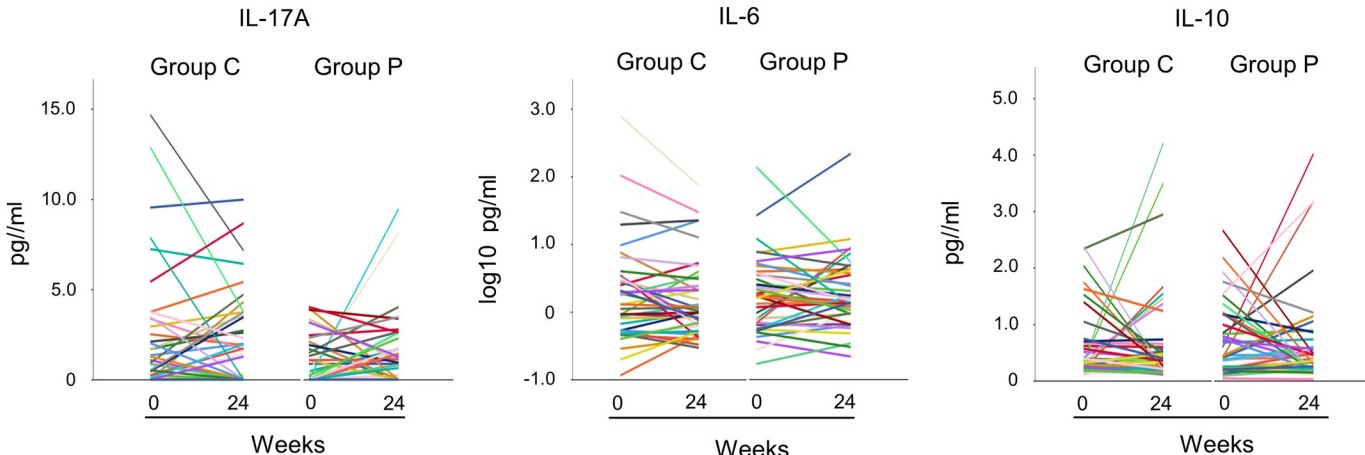

| Change in each cytokine category from 0 to 24 weeks (ANCOVA) | | |
|---|---|---|
| | Item | Effect (95%CI) |
| Group P (vs Group C) | IL17-A | -0.11 (-0.90, 0.73) |
| | IL-6 | 4.59 (-2.40, 17.20) |
| | IL-10 | -0.04 (-0.46, 0.35) |

**Fig 4. Changes in cytokines over the course of the study in the FAS.** Bootstrap confidence intervals are calculated because the normality of the residuals is problematic (repeated 1000 times). The IL-6 value is shown as a log10 value due to the large variability of values. ANCOVA; analysis of covariance, C, control; CI, confidence interval; IL, interleukin; P, propolis.

on suppression of disease activity or improvement of QOL was found for patients with RA. Okamoto et al. found that in a mouse model of RA, propolis contributed to suppression of disease activity in RA. The contributing mechanism was reported to involve propolis inhibition of the phosphorylation of STAT3 and subsequent suppression of IL-17 production [11]. However, IL-17A of the IL-17 family was measured in our study, and no significant reduction was apparent with propolis intake. Since IL-17 induces the expression of pro-inflammatory cytokines such as IL-6 and tumor necrosis factor-α in cells such as macrophages [23], we also measured IL-6. As with IL-17A, no significant decrease was seen following propolis intake. Since pro-inflammatory cytokines are strongly associated with disease activity in RA, propolis was assumed to exert no suppressive effects on disease activity. Since disease activity affects QOL, no effects on QOL were identified because of the lack of findings on suppression of disease activity. A similar interpretation could be made with regard to joint sonographic assessments and ADLs. In this study, reasons for the null finding of effects of propolis on the suppression of disease activity in RA patients may be that the participating patients were already receiving pharmacotherapy under a strategy of treat-to-target [4]. It A medication might be considered to exert strong effects on disease activity, propolis may not have shown suppressive effects on disease activity that surpassed drug effects. In addition, in our study, baseline IL17A levels were similar to those reported in other RA patients [24], but levels of IL-6 were lower compared to another study even at the same level of disease activity [25]. Therefore, testing in populations with higher cytokine levels might thus be warranted. In addition, since diets such as vegetarian diets [26], Mediterranean diets [21], and anti-inflammatory diets [27] have been reported to suppress disease activity in RA patients, examination of add-on effects of propolis

**Table 2. Adverse events over the course of the study in the SAS.**

| Adverse event | Group C | Group P |
|---|---|---|
| | (n = 13) | (n = 11) |
| Common cold | 4 (30.7) | 1 (9.1) |
| Influenza | 0 (0) | 1 (9.1) |
| Pneumonia | 1 (7.7) | 0 (0) |
| Cough | 1 (7.7) | 0 (0) |
| Gastroenteritis | 1 (7.7) | 1 (9.1) |
| Bacteriogenic enteritis | 1 (7.7) | 0 (0) |
| Herpes | 1 (7.7) | 1 (9.1) |
| Eruption | 0 (0) | 1 (9.1) |
| Fall | 0 (0) | 1 (9.1) |
| Sprain (shoulder) | 1 (7.7) | 0 (0) |
| Fracture | 0 (0) | 3 (27.3) |
| Gonitis | 0 (0) | 1 (7.1) |
| Acute low back pain | 1 (7.7) | 0 (0) |
| Spinal canal stenosis and worsening back pain | 1 (7.7) | 0 (0) |
| Carpal tunnel syndrome | 0 (0) | 1 (9.1) |
| Cramp | 0 (0) | 1 (9.1) |
| Increased blood levels of tacrolimus | 0 (0) | 1 (9.1) |
| Anemia | 2 (15.4) | 0 (0) |
| Nail loss (foot) | 1 (7.7) | 0 (0) |
| Weight gain | 1 (7.7) | 0 (0) |

Data are shown as number of patients (%).

SAS, safety analysis set; C, control; P, propolis.

in addition to these dietary treatments on disease activity in patients with RA may be necessary.

In this study, we reported the status of adverse events in RA patients treated with propolis as an intervention study. One serious adverse event, a fracture, was observed in the propolis-treated group, but the relationship between the fracture and propolis treatment was unclear. No characteristic increase in all adverse events with propolis administration was identified during the conduct of this study. Although no inhibitory effect of propolis on disease activity of RA was observed in our study, since beneficial reports (such as preventive effects of propolis on cognitive decline) have been reported [9], our data suggest that propolis could serve as a resource for RA patients who expect these functions to take propolis under the guidance of their physicians.

Our study was a double-blind, randomized controlled trial, but some limitations should be considered. First, the reason for including patients with MDA in this study was that patients with high disease activity require more intensive medication. Propolis is not a drug and was considered unlikely to be as powerful as regular RA treatments. However, even with MDA, patients did not reach efficacy targets, and one possibility is that this group might still have needed intense treatment. Other perspectives on examining the effect of propolis on disease activity in RA might be needed, such as examining effects on the duration of remission or low disease activity. We need to have an improved power calculation based on more realistic effect sizes in a representative population. Second, we used only one commercially available dose with a dosing period of 6 months. Different results might be obtained with higher doses or longer dosing periods. Third, this study was conducted only on women. It would be desirable to verify the effect of propolis on male patients with RA in the future.

## Conclusions

No effect of Brazilian propolis administration was seen on disease activity suppression in RA patients with moderate disease activity.

## Supporting information

**S1 Fig. Changes in DAS28-CRP over the course of the study in the FAS.** Data are presented as box-and-whisker plots. White box, Group C; gray box, Group P. P values are calculated using a mixed-effects model. C, control; CI, confidence interval; CRP, C-reactive protein; DAS, disease activity score; P, propolis.
(TIF)

**S2 Fig. Changes in SDAI and CDAI over the course of the study in the FAS.** Data are presented as box-and-whisker plots. White box, Group C; gray box, Group P. Bootstrap confidence intervals are calculated because the normality of the residuals is problematic (repeated 1000 times). C, control; CDAI, clinical disease activity index; CI, confidence interval; SDAI, simplified disease activity index; P, propolis.
(TIF)

**S3 Fig. Changes in total joint sonographic scores in 7 joints over the course of the study in the FAS.** Data are presented as box-and-whisker plots. White box, Group C; gray box, Group P. Because the normality of residuals in the model was problematic, statistical analysis is performed after log-transforming total joint sonographic scores at each time point. P values are calculated using a mixed-effects model. C, control; CI, confidence interval; P, propolis.
(TIF)

**S4 Fig. Changes in mHAQ score over the course of study in the FAS.** Data are presented as box-and-whisker plots. White box, Group C; gray box, Group P. Bootstrap confidence intervals are calculated because the normality of the model's residuals is problematic (repeated 1000 times). C, control; CI, confidence interval; mHAQ, modified health assessment questionnaire disability index; P, propolis.
(TIF)

**S1 File. Data set.**
(CSV)

**S2 File.**
(DOCX)

**S3 File.**
(PDF)

**S4 File.**
(DOC)

## Acknowledgments

We wish to thank Osaka City University Hospital Center for Clinical Research and Innovation and Atsuko Kamiyama for collecting and managing the data, Tomoko Nakatsuka for monitoring the study, and all study subjects for their participation.

## Author Contributions

**Conceptualization:** Yoshinari Matsumoto, Kentaro Inui, Ayumi Shintani, Tatsuya Koike.

**Formal analysis:** Yoshinari Matsumoto, Kanae Takahashi, Ayumi Shintani.

**Funding acquisition:** Yoshinari Matsumoto.

**Investigation:** Yuko Sugioka, Kentaro Inui, Tadashi Okano, Koji Mandai, Yutaro Yamada, Tatsuya Koike.

**Methodology:** Yoshinari Matsumoto, Kanae Takahashi, Yuko Sugioka, Kentaro Inui, Ayumi Shintani, Tatsuya Koike.

**Project administration:** Tatsuya Koike.

**Resources:** Yoshinari Matsumoto, Yuko Sugioka, Kentaro Inui, Tadashi Okano, Koji Mandai, Yutaro Yamada, Tatsuya Koike.

**Supervision:** Tatsuya Koike.

**Visualization:** Yoshinari Matsumoto, Kanae Takahashi, Tatsuya Koike.

**Writing – original draft:** Yoshinari Matsumoto, Tatsuya Koike.

**Writing – review & editing:** Yoshinari Matsumoto, Ayumi Shintani, Tatsuya Koike.

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
