## [Decision Letter · Decision Letter 0]

9 Mar 2021

PONE-D-20-39470

Double-blinded randomized controlled trial to reveal the effects of Brazilian propolis intake on rheumatoid arthritis disease activity index; BeeDAI

PLOS ONE

Dear Dr. Koike,

Thank you for submitting your manuscript to PLOS ONE. After careful consideration, we feel that it has merit but does not fully meet PLOS ONE’s publication criteria as it currently stands. Therefore, we invite you to submit a revised version of the manuscript that addresses the points raised during the review process.

We look forward to receiving your revised manuscript.

Kind regards,

Yuanyuan Wang, PhD

Academic Editor

PLOS ONE

Additional Editor Comments:

The reviewers have raised methodological issues, in particular statistical analyses. The authors will need to appropriately address these issues to make the manuscript scientifically sound.

Journal Requirements:

2. Thank you for submitting your clinical trial to PLOS ONE and for providing the name of the registry and the registration number. The information in the registry entry suggests that your trial was registered after patient recruitment began. PLOS ONE strongly encourages authors to register all trials before recruiting the first participant in a study.

1) your reasons for your delay in registering this study (after enrolment of participants started);

2) confirmation that all related trials are registered by stating: “The authors confirm that all ongoing and related trials for this drug/intervention are registered”.

"YM received Yamada Research Grant

Specific grant numbers

Yamada Research Grant (grant No.249)

Full names of commercial companies that funded the study or authors

Yamada Bee Company, Inc.

Initials of authors who received salary or other funding from commercial companies

There were no authors.

URLs to funder websites

https://www.3838.com

Also state whether any sponsors or funders (other than the named authors) played any role in: YES

・Preparing and blinding propolis and placebo tablets

・Decision to publish"

We note that you received funding from a commercial source: Yamada Bee Company, Inc.

"Yoshinari Matsumoto; Grant/research support from: Yamada Research Grant

Kanae Takahashi; Grant/research support/ Speakers bureau: no COI

Kentaro Inui; Grant/research support: Janssen Pharmaceutical K.K., Astellas Pharma Inc., Sanofi K.K., Abbvie GK, Takeda Pharmaceutical Co. Ltd., QOL RD Co. Ltd., Mitsubishi Tanabe Pharma, Ono Pharmaceutical Co. Ltd., Eisai Co., Ltd., Speakers bureau: Daiichi Sankyo Co. Ltd., Mitsubishi Tanabe Pharma, Janssen Pharmaceutical K.K., Astellas Pharma Inc., Takeda Pharmaceutical Co. Ltd., Ono Pharmaceutical Co. Ltd., Abbvie GK, Pfizer Inc., Eisai Co., Ltd., Chugai Pharmaceutical Co., Ltd.,

Tadashi Okano; Grant/research support: AbbVie, Eisai, Mitsubishi Tanabe Pharma Corporation and Nipponkayaku, Speakers bureau: AbbVie, Asahikasei, Astellas Pharma Inc, Ayumi Pharmaceutical, Bristol-Myers Squibb, Chugai Pharmaceutical, Daiich Sankyo, Eisai, Janssen, Lilly, Mitsubishi Tanabe Pharma Corporation, Novartis Pharma, Ono Pharmaceutical, Pfizer, Sanofi, Takeda Pharmaceutical, Teijin Pharma and UCB

Koji Mandai; Speakers bureau: Asahikasei, Eisai

Yutaro Yamada; Speakers bureau: Abbvie, Chugai, Eisai, and Mitsubishi Tanabe

Ayumi Shintani: Grant/research support: no COI, Speakers bureau: Chugai Pharmaceutical Co., Ltd.

Tatsuya Koike; Grant/research support: AbbVie, Astellas Pharma Inc, Bristol-Myers Squibb, Chugai Pharmaceutical, Eisai, Janssen, Lilly, Mitsubishi Tanabe Pharma Corporation, MSD, Ono Pharmaceutical, Pfizer, Roche, Takeda Pharmaceutical, Teijin Pharma, and UCB, Speakers bureau: AbbVie, Astellas Pharma Inc, Bristol-Myers Squibb, Chugai Pharmaceutical, Eisai, Janssen, Lilly, Mitsubishi Tanabe Pharma Corporation, MSD, Ono Pharmaceutical, Pfizer, Roche, Takeda Pharmaceutical, Teijin Pharma, and UCB. This study has not received specific funding from any source.

This research was supported (in part) by Yamada Research Grant."

Reviewers' comments:

Reviewer's Responses to Questions

**Comments to the Author**

1. Is the manuscript technically sound, and do the data support the conclusions?

Reviewer #1: Yes

Reviewer #2: Yes

Reviewer #3: Yes

2. Has the statistical analysis been performed appropriately and rigorously? 

Reviewer #1: Yes

Reviewer #2: No

Reviewer #3: No

3. Have the authors made all data underlying the findings in their manuscript fully available?

Reviewer #1: No

Reviewer #2: No

Reviewer #3: Yes

4. Is the manuscript presented in an intelligible fashion and written in standard English?

Reviewer #1: Yes

Reviewer #2: Yes

Reviewer #3: Yes

5. Review Comments to the Author

Reviewer #1: Dear authors,

Thank you for this well written and thoroughly designed study on the effect of propolis on disease activity in RA. i have to admit I did not know this substance, so first of all thank you for this. Although this is a negative trial, eg the effect is missing, I believe this study to be sound and deserves publication. However, before this is warranted, a few questions should be answered

- In this RCT only women were included. Why?

- Moderate disease activity is perhaps not an ideal state to discover very small effects in inflammation changes. One could also think of inhibition of flares, or as adjuvans in early RA with intensive treatment strategies. However, doing this in practice is perhaps a lost effort as this trial clearly shows no benefit at all of the supplements.

- For the power calculation, there is an effect size of 0.5 assumed. Perhaps this was too ambitious and more persons were needed to detect very small effect sizes... I would write such a statement in limitations.

- I now have mastered all knowledge on propolis via Wikipedia and have read that propolis is very heterogeneous in content. Could it be that one specific type of propolis does work better than the other? Why was the Brazilian propolis chosen?

Good luck!

Reviewer #2: A two-arm randomized placebo-controlled clinical trial (n=80) was conducted to compare the effects of Brazilian propolis to placebo on change in disease activity in RA patients. No significant differences were observed in Disease Activity Score or any secondary endpoints between the two arms.

Minor revisions:

1- Line 226: Consider editing the phrase “If no normality was observed...”

2- Line 230: Indicate the underlying covariance structure used in the mixed-effects models and the criteria for selecting it.

3- Line 252: Provide a measure of dispersion (range or interquartile range) for the median age.

4- The abstract indicates that analysis of covariance methods were used to analyze the data; the statistical analysis section, however, states that mixed-effects models were used. Please clarify. The statistical analysis section should include a comprehensive listing and explanation of all the statistical methods used for the analysis.

5- The study is likely underpowered to test the interaction of arm by time. Plus the sample size justification/power calculation is based upon a t-test, not a test of the interaction effect. Please edit the manuscript for clarification. If tests of interaction effects were conducted, the results have not been presented. If tests of interactions are conducted indicate if the following procedure was implemented when testing interaction effects and main effects. If the interaction effect is significant, provide an interpretation of the results, but do not test main effects because the tests for main effects are uninteresting in light of significant interactions. If interaction effects are non-significant, drop the interaction effects from the model and test the main effects. Determining which results to present when testing interactions is often a multi-step process.

6- When presenting results from mixed effect models, please clarify what, if any, other variables than those listed in the tables were included in the model.

7- Indicate the date range subjects were enrolled in the study.

Reviewer #3: Line 39: please elaborate on the term moderate disease activity, was that assessed by use of DAS28-CRP, or?

Line 42: the abstract describes the use of propolis tablets containing 508.5 mg of propolis whereas the trial registration describes the use of tablets containing 378 mg propolis, please clarify (https://rctportal.niph.go.jp/en/detail?trial_id=jRCTs051180142)

Line 43: were the tablets similar in shape and color?

Line 204: I would recommend including an analysis of the ITT population, defined as every patient randomized as that would minimize a potential bias due to exclusion of patients. Moreover, the ITT analysis reflects the practical clinical scenario because it admits noncompliance and protocol deviations, pls see https://www.ncbi.nlm.nih.gov/pmc/articles/PMC3159210/.

Line 207: I would recommend adapting the FAS population to all patients randomly assigned to a treatment group having at least one efficacy assessment after randomization and participants who never consumed the test food.

As highlighted by EMA (https://www.ema.europa.eu/en/documents/scientific-guideline/ich-e-9-statistical-principles-clinical-trials-step-5_en.pdf) there a only very few circumstances that may lead to the exclusion of randomized subjects in an FAS analysis (one circumstance already being mentioned in the manuscript, “participants who never consumed the test food”).

If the criteria “participants found to be ineligible after randomization” is to be used this paragraph must be expanded (pls see link above).

Line 218: I would recommend defining the Safety Analysis Set as all randomized patients with at least one safety assessment after randomization.

Line 230: why include week 12 data in the analysis of the primary endpoint?

Table 1: did none of the patients receive csDMARDS of ts-DMARDs?

Figure 1: needs to be updated to include i.e. reasons for discontinued intervention (box-row 5 from the top, pls see https://www.equator-network.org/reporting-guidelines/consort/)

6. PLOS authors have the option to publish the peer review history of their article (what does this mean?). If published, this will include your full peer review and any attached files.

Reviewer #1: **Yes: **Diederik De Cock

Reviewer #2: No

Reviewer #3: No

---

## [Author Response · Author response to Decision Letter 0]

4 Apr 2021

Please see Response to Reviewers letter at the end of this PDF.

---

## [Decision Letter · Decision Letter 1]

4 May 2021

PONE-D-20-39470R1

Double-blinded randomized controlled trial to reveal the effects of Brazilian propolis intake on rheumatoid arthritis disease activity index; BeeDAI

PLOS ONE

Dear Dr. Koike,

Thank you for submitting your manuscript to PLOS ONE. After careful consideration, we feel that it has merit but does not fully meet PLOS ONE’s publication criteria as it currently stands. Therefore, we invite you to submit a revised version of the manuscript that addresses the points raised during the review process.

We look forward to receiving your revised manuscript.

Kind regards,

Yuanyuan Wang, PhD

Academic Editor

PLOS ONE

Journal Requirements:

Additional Editor Comments (if provided):

The reviewers have made additional comments that will help improve the manuscript.

Reviewers' comments:

Reviewer's Responses to Questions

**Comments to the Author**

1. If the authors have adequately addressed your comments raised in a previous round of review and you feel that this manuscript is now acceptable for publication, you may indicate that here to bypass the “Comments to the Author” section, enter your conflict of interest statement in the “Confidential to Editor” section, and submit your "Accept" recommendation.

Reviewer #1: (No Response)

Reviewer #2: All comments have been addressed

Reviewer #3: All comments have been addressed

2. Is the manuscript technically sound, and do the data support the conclusions?

Reviewer #1: Yes

Reviewer #2: (No Response)

Reviewer #3: Yes

3. Has the statistical analysis been performed appropriately and rigorously? 

Reviewer #1: Yes

Reviewer #2: (No Response)

Reviewer #3: No

4. Have the authors made all data underlying the findings in their manuscript fully available?

Reviewer #1: No

Reviewer #2: (No Response)

Reviewer #3: Yes

5. Is the manuscript presented in an intelligible fashion and written in standard English?

Reviewer #1: Yes

Reviewer #2: (No Response)

Reviewer #3: Yes

6. Review Comments to the Author

Reviewer #1: Dear authors,

Please put in limitations that

1. These results only apply to women, and that studies in men should be considered.

2. Refrase line 324-326: "We might have overestimated the effectiveness of propolis and might need to develop clinical studies to verify less efficacy in the future." You dont want to verify less efficacy, you want to make sure that propolis has or do not has an effect. Therefore, you need to have an improved power calculation based on more realistic effect sizes in a representative population.

Reviewer #2: (No Response)

Reviewer #3: Thanks for your replies to the comments provided

As previously pointed out I would strongly urge you to adapt the description of your FAS population according to ICH as there are a limited number of circumstances that might lead to excluding randomized subjects from the full analysis set including

- the failure to satisfy major entry criteria (eligibility violations),

- the failure to take at least one dose of trial medication and

- the lack of any data post randomisation

The trial figure is to be updated according to the CONSORT guideline (https://www.equator-network.org/reporting-guidelines/consort/)

The manuscript describes that in Group C the study was discontinued in 4 cases, this is not reflected in figure 1

7. PLOS authors have the option to publish the peer review history of their article (what does this mean?). If published, this will include your full peer review and any attached files.

Reviewer #1: **Yes: **Diederik De Cock

Reviewer #2: No

Reviewer #3: **Yes: **Henrik Gudbergsen

---

## [Author Response · Author response to Decision Letter 1]

7 May 2021

Dear Editor and Reviewers;

We would like to thank you for your very constructive criticisms and editorial suggestions regarding our manuscript. These were very helpful in our sincere efforts to improve our manuscript. We addressed all the concerns point-by-point, depicting Reponses by use of bold font, and revised the manuscript accordingly. We hope that our response meets with your approval. We indicate revised wording with yellow highlighting in the text.

Reviewer #1: Dear authors,

Please put in limitations that

1. These results only apply to women, and that studies in men should be considered.

Response: Thank you for your comment. We have added a note about the need for studies with men. We added a sentence in the text, “Third, this study was conducted only on women. It would be desirable to verify the effect of propolis on male patients with RA in the future.”

2. Refrase line 324-326: "We might have overestimated the effectiveness of propolis and might need to develop clinical studies to verify less efficacy in the future." You dont want to verify less efficacy, you want to make sure that propolis has or do not has an effect. Therefore, you need to have an improved power calculation based on more realistic effect sizes in a representative population.

Response: Thank you for your comment. We have corrected the description in this section. We changed the sentence to “We need to have an improved power calculation based on more realistic effect sizes in a representative population.”.

Reviewer #3: Thanks for your replies to the comments provided

As previously pointed out I would strongly urge you to adapt the description of your FAS population according to ICH as there are a limited number of circumstances that might lead to excluding randomized subjects from the full analysis set including

- the failure to satisfy major entry criteria (eligibility violations),

- the failure to take at least one dose of trial medication and

- the lack of any data post randomisation

Response: Thank you for your advice. Actually, no participant violated the above conditions, but we added 2 items to the definition part of FAS as 4) and 5). Participants who have never taken the test tablet have already been mentioned.

The trial figure is to be updated according to the CONSORT guideline (https://www.equator-network.org/reporting-guidelines/consort/)

Response: Thank you for pointing this out, we have modified the flowchart in Fig.1 to match the CONSORT diagram.

The manuscript describes that in Group C the study was discontinued in 4 cases, this is not reflected in figure 1

Response: Thank you for pointing out our misleading statement. No one in the control group failed to complete the test. It was correct that the number of discontinued intervention is zero. At the final check stage, four subjects took less than 70% of test tablets. The description in the text and Figure 1 have been revised to make it easier to understand.

---

## [Editor Report · Decision Letter 2]

14 May 2021

Double-blinded randomized controlled trial to reveal the effects of Brazilian propolis intake on rheumatoid arthritis disease activity index; BeeDAI

PONE-D-20-39470R2

Dear Dr. Koike,

We’re pleased to inform you that your manuscript has been judged scientifically suitable for publication and will be formally accepted for publication once it meets all outstanding technical requirements.

Kind regards,

Yuanyuan Wang, PhD

Academic Editor

PLOS ONE
---

## [Editor Report · Acceptance letter]

20 May 2021

PONE-D-20-39470R2 

Double-blinded randomized controlled trial to reveal the effects of Brazilian propolis intake on rheumatoid arthritis disease activity index; BeeDAI 

Dear Dr. Koike:

I'm pleased to inform you that your manuscript has been deemed suitable for publication in PLOS ONE. Congratulations! Your manuscript is now with our production department. 

Kind regards, 

on behalf of

Dr. Yuanyuan Wang 

Academic Editor

PLOS ONE